# Gender and urban–rural residency based differences in the prevalence of type-2 diabetes mellitus and its determinants among adults in Naghadeh: Results of IraPEN survey

Nafiseh Ghassab-Abdollahi[1], Haidar Nadrian[2]*, Kobra Pishbin[3], Shayesteh Shirzadi[4], Parvin Sarbakhsh[5], Fatemeh Saadati[6], Mohammad Sanyar Moradi[7], Pouria Sefidmooye Azar[8], Leila Zhianfar[9]

1 Faculty of Health, Department of Health Education and Promotion, Aging Research Institute, Student Research Committee, Ph.D. Student in Gerontology, Tabriz University of Medical Sciences, Tabriz, Iran, 2 Faculty of Health, Department of Health Education, Tabriz University of Medical Sciences, Tabriz, Iran, 3 Faculty of Health, Department of Health Education and Promotion, M.Sc. in Community-Based Education in Health System, Tabriz University of Medical Sciences, Tabriz, Iran, 4 Public Health Department, Neyshabur University of Medical Sciences, Neyshabur, Iran, 5 Faculty of Health, Department of Epidemiology and Biostatistics, Tabriz University of Medical Sciences, Tabriz, Iran, 6 Department of Health Education & Promotion, School of Health, Student Research Committee, PhD Student in Health Education and Promotion, Tabriz University of Medical Sciences, Tabriz, Iran, 7 Department of Occupational Health, B. S. Student in Occupational Health, School of Health, Shiraz University of Medical Sciences, Shiraz, Iran, 8 Ph.D. Students in Nutrition Sciences, The University of Mississippi, University, Mississipp, United States of America, 9 Assistant Professor in Community Health Department, School of Nursing & Midwifery, Mazandaran University of Medical Sciences, Sari, Iran

* haidarnadrian@gmail.com

## Abstract

### Background

Type 2 diabetes mellitus (T2DM) is one of the most important risk factors for cardiovascular diseases, with a high economic burden on health care systems. Since gender and residency can affect people's lifestyle and health behaviors, this study was conducted to investigate the prevalence of T2DM and its determinants by gender and residency.

### Methods

A secondary analysis study was conducted on the survey data of the IraPEN (Iran's Package of Essential Non-Communicable Disease) pilot program conducted in 2017 in Naghadeh County, Iran. Data of 3,691 participants aged 30–70 years from rural and urban areas of the County were included into data analysis process. Sociodemographic factors, anthropometric measurements, and cardiovascular risk factors related to T2DM were assessed.

### Results

The overall prevalence of T2DM within the population was 13.8%, which was significantly higher among women (15.5%) than men (11.8%), and non-significantly higher in urban (14.5%) areas than rural (12.3%) areas. In both genders, age (male: OR 1.01, 95% CI:

**Data Availability Statement:** All relevant data are uploaded to Figshare: https://doi.org/10.6084/m9.figshare.20293767.v1.

**Funding:** The authors received no specific funding for this work.

**Competing interests:** The authors have declared that no competing interests exist.

1.00–1.03; P = 0.012; female: OR 1.03, 95% CI: 1.02–1.04; P<0.001), blood pressure (male: OR 1.77, 95% CI: 1.13–2.79; P = 0.013; female: OR 2.86, 95% CI: 2.12–3.85; P<0.001), and blood triglycerides (male: OR 1.46, 95% CI: 1.01–2.11; P = 0.04; female: OR 1.34, 95% CI: 1.02–1.77; P = 0.035) had a significant relationship with the chance of developing T2DM. Among women, a significant relationship was found between abdominal obesity (OR 1.68, 95% CI: 1.17–2.40; P = 0.004) and the chance of developing T2DM. Age (rural: OR 1.03, 95% CI: 1.01–1.04; P<0.001; urban: OR 1.02, 95% CI: 1.01–1.04; P<0.001), blood pressure (rural: OR 3.14, 95% CI: 2.0–4.93; P<0.001; urban: OR 2.23, 95% CI: 1.66–3; P<0.001), and abdominal obesity (rural: OR 2.34, 95% CI: 1.41–3.87; P = 0.001; urban: OR 1.46, 95% CI: 1.06–2.01; P = 0.019), in both rural and urban areas, blood cholesterol (OR 1.59, 95% CI: 1.07–2.37; P = 0.02) in rural areas, and blood triglycerides (OR 1.51, 95% CI: 1.16–1.98; P = 0.002) in urban areas were significant predictors of T2DM.

## Conclusion

Given the higher prevalence of T2DM among females, risk reduction strategies at the community level should be more targeted at women. The higher prevalence of T2DM risk factors among the urban population is a wake-up call for policymakers to pay more attention to the consequences of unhealthy and sedentary lifestyles within urban communities. Future actions should be focused on appropriate timely action plans for the prevention and control of T2DM from early years of life.

## Introduction

Type 2 diabetes mellitus (T2DM) is a complex disorder that results from several pathophysiological complications including decreased insulin secretion, increased glucose production in the liver, and increased insulin resistance [1]. T2DM is the most common type of diabetes and approximately accounts for 90 to 95% of all diagnosed diabetes cases [2]. The number of people aged over 18 years with T2DM in 2014 was 422 million, which was equivalent to a prevalence of about %8.5, worldwide. The highest prevalence of T2DM occurs in middle and low-income countries. The prevalence of T2DM is continuously increasing in these countries [3]. In Iran, as a developing country. the prevalence of diabetes in the population older than 40 is more than 24% [4].

The lack of definitive treatment and its deadly effects have made T2DM one of the most challenging diseases. The disease is also one of the most important risk factors for cardiovascular diseases, and the most common cause of microvascular complications, such as amputation, blindness, and chronic renal failure, which can all affect the patients' quality of life [1]. Such complications impose a heavy economic burden to health care systems and society, as well [5]. In Iran, diabetes was the sixth-leading cause of death among the population in 2014 [6].

Various studies have identified several risk factors associated with T2DM, such as a family history of diabetes, obesity, age over 45 years, race, history of gestational diabetes, high blood pressure, high cholesterol, low HDL, high LDL, and glucose tolerance disorder [7]. Among them, sociodemographic and lifestyle-related factors were identified to be related to high T2DM risk [8]. Obesity, smoking, alcohol consumption, high blood pressure, and dyslipidemia, as consequences of a sedentary lifestyle, are among the important unhealthy conditions and behaviors that may increase the risk of T2DM [9,10].

Gender differences in health are associated to the difference in lifestyle factors and context [11]. Differences in health-protective behaviors between males and females have previously been reported [12]. Men's health was more influenced by health behaviors while women's health was more affected by structural determinants [11]. Gender differences in the prevalence of diabetes have also been reported in the literature [13], and some aspects of gender were identified as T2DM risk factors [14].

In addition to gender, urbanization affects people's lifestyles and socioeconomic conditions. Diversity in the demographic variables, health behaviors, and lifestyle are effective factors in the existence of differences in the prevalence of diabetes among urban and rural areas. Urbanization is associated to better access to health, education, and social services. On the other hand, adverse changes in health behaviors and western lifestyle among people in urban areas have led to an increase in the rates of obesity, which can be associated to the rates of T2DM [15–17].

Complex pathophysiological processes resulting from interactions between genes and environment suggest that the T2DM risk factors can vary within different populations. Conventional risk factors for predicting diabetes vary among countries and geographical regions. This difference in the prevalence of risk factors may result from population structure such as aging, cultural context, and lifestyle factors, such as diet and physical activity [8,18].

So far, some studies have identified gender and residency differences in the prevalence of diabetes [13–15]. Due to the considerable socio-cultural and environmental diversity within and across different countries, identifying important risk factors of non-communicable diseases (NCDs) in each community may be helpful in preventing adverse outcomes, and improving population's health [7]. World Health Organization (WHO) recommends identifying and collecting data that help to demonstrate the impact of cultural differences on health [19]. Particularly, prioritizing the lifestyle-related risk factors of T2DM by gender and residency may provide health policymakers with different strategies in the prevention and control of the disease. The present study aimed to investigate the prevalence of T2DM and its determinants in the population older than 30 in Naghadeh-Iran, based on gender and urban-rural differences using IraPEN data.

## Materials and methods

### About IraPEN program in Naghadeh

The WHO Package of Essential Noncommunicable Disease Interventions (WHO PEN) for primary care in low-resource settings is an innovative and action-oriented set of cost-effective interventions that might not only reduce medical costs, but also increase the patient's quality of life [20]. WHO has defined the control of NCDs and their underlying factors as the main goal to reduce their associated mortality by 2025. IraPEN (Iran's Package of Essential Non-Communicable Disease) is a prioritized group of effective interventions that is part of the national health transformation plan created by the Iranian Ministry of Health and Medical Education, in 2014. IraPEN is a response of the Iranian health system to the long-term goals of WHO, with the hope to prevent the four main types of NCDs (diabetes, cancer, respiratory and cardiovascular diseases), and to reduce their associated risk factors among 30 to 70 years old population. The project was based on the WHO PEN, which was adjusted for Iranian setting, and pilot-tested in four counties; Naghadeh, Maragheh, Shahreza, and Baft. IraPEN was then integrated into the primary healthcare services at the health centers, nationwide.

The implementation of this project in the health centers of Naghadeh was carried out according to the *"Package of essential noncommunicable diseases in Iran's primary health care system"* developed by the Ministry of health, under the supervision of the representative office

of WHO in Iran [21]. The initial goal of IraPEN was to estimate the prevalence of four conditions (diabetes, high blood pressure, cancer, and chronic respiratory diseases) along with their risk factors: sedentary lifestyle, unhealthy diet, alcohol consumption, and smoking [22–24].

Naghadeh County is located in West Azerbaijan province, the northwest of Iran. Based on the last Iranian census in 2016, the population of Naghaded was 127671 people, and features two Turk and Kurd ethnic groups with almost equal population [25]. The population of 30 to 70 years old men and women in urban and rural areas were 30049 and 29075, and 8403 and 8024, respectively.

## Study design and setting

This study was a secondary analysis study conducted on the survey data of the IraPEN pilot project. IraPEN launched in the County in 2017. We accessed the project data in 2018. The participants were recruited using their household health records in the health centers of both urban and rural areas. Rural area is defined as a land with a specific area and territory, where either at least 20 households or 100 people live. An urban area is also defined as the region in which most inhabitants have nonagricultural jobs. It has a population of at least ten thousand people [26]. In urban areas of Naghadeh, the major economic activities are self-employment, governmental employment, and the trade of agricultural/animal husbandry-associated products. In the rural areas, a majority of residents are directly/indirectly engaged in one of the activities of agriculture, animal husbandry, horticulture, and rural industries. All people aged 30–70 years old who received the care related to the IraPEN project in the health centers, and had data in the project were eligible to be included into our secondary analysis.

## Data collection and measurements

The process of data collection was based on the IraPEN guideline, namely the "Package of essential noncommunicable diseases in Iran's primary health care system" [21], which was the directive of the Ministry of Health. The IraPEN standard forms were used to collect data. IraPEN form had two sections. In the first section, all participants responded to sociodemographic questions such as age, gender, ethnicity, marital status, education levels, occupation, history of smoking and alcohol consumption, history of hypertension, and history of diabetes in the first-degree relatives. The second section of the form included 1) anthropometric measurements such as weight, height, body mass index (BMI), waist circumference (WC), hip circumference, waist to hip ratio (WHR), and 2) cardiovascular risk factors such as fasting blood sugar, blood lipids (cholesterol, triglyceride), and systolic and diastolic blood pressure (BP). All forms were completed by trained health care providers.

A calibrated scale to the nearest 0.1 kg was used to weigh an individual with light clothing without shoes. A stadiometer to the nearest 0.1 cm was also used to measure height without shoes. BMI was calculated by a participant's weight in kilograms divided by the square of the height in meters. The BMIs less than 25, from 25 to 29.9, and above 30 were considered to be under/normal weight, overweight and obese, respectively. WC was measured by placing a tape measure around the waist horizontally above the hipbones. Hip circumference was also measured by a tape measure to check around the largest/widest part of the buttocks. WHR was also calculated by dividing waist to hip circumference. WHRs above 0.85 in women and above 0.9 in men were considered as abdominal obesity [27].

Participant's BP was measured with a mercury sphygmomanometer twice with an interval of at least 5 minutes. To measure BP, digital sphygmomanometer is recommended by WHO [28]. However, mercury sphygmomanometer was used in the IraPEN study, considering that digital blood pressure measuring device was not available in all rural and urban health centers

of the country. The average blood pressure of two measurements higher than 140/90 mmHg was considered to be high blood pressure [21]. In IraPEN, the status of smoking and alcohol consumption were assessed using two separate modified items derived from the Alcohol, Smoking and Substance Involvement Screening Test (ASSIST). The practical definition of smoking and alcohol consumption in IraPEN guideline was defined as follow: "In the past three months, have you smoked cigarette and/or hookah (even for one time)?" and "In the past three months, have you had alcohol consumption (even for one time)?" [21].

Blood sampling was collected from all participants to measure glycemic and lipid parameters. Venous blood samples (5 ml.) were collected. Before blood sampling, 8 hours of fasting was needed. In IraPEN, Lipid pro® was used to assess blood glucose and lipids. The instructions for using this tool were provided in detail in the IraPEN guideline for health workers [21].

The glucose oxidase method (intra- and inter-assay coefficients of variation 2.1% and 2.6%, respectively) was used to measure fasting blood sugar (FBS). T2DM was defined as the fasting plasma glucose level higher than 126 mg/dl ($\geq$7.0 mmol/L), confirmed by a physician [29]. In cases where FBS was above 126 mg/dl in the first measurement, the blood test was repeated, immediately. If FBS was again 126 mg/dl and/or higher in the second measurement, the participant was diagnosed as a diabetic patient. The lipid profile test was also taken in a fasting condition.

Oxidase methods were used for measuring fasting cholesterol and triglyceride levels in plasma. Plasma cholesterol levels more than 200 (mg/dL) and triglyceride levels more than 150 (mg/dL) were considered as high blood cholesterol and high triglyceride, respectively [21].

## Statistical analysis

Data were analyzed using Statistical Package for Social Science (SPSS 23 for windows, SPSS Inc.® headquarter, Chicago, USA). The Kolmogorov–Smirnov test was applied to assess the normality of data. Descriptive statistics including frequency and percentage were employed to express the data. A Chi-square test was used to compare variables by gender and residency.

Simple and multiple logistic regression models were used to examine the odds of developing T2DM based on the risk factors by gender and residency. Considering each risk factor as an independent variable, and gender and residency as dependent variables, a series of multiple logistic regression analysis with stepwise forward variable selection method was used. In this model, we moved from a simpler to a more complex model to understand the contribution of variables. The variables with p < .05 in the simple regression analysis were included in the multiple models. Adjusted odds ratio (OR) of T2DM risk factors with 95% confidence intervals were provided. The p-values less than 0.05 were considered to be statistically significant.

## Ethical considerations

The ethics committee of Tabriz University of Medical Sciences approved the study protocol (IRTBZMED.REC.1396.965). When data collection, a signed informed consent form was obtained from all participants.

## Results

### Overall prevalence of T2DM by gender, residency and socio-demographic characteristics

The records of 3,691 participants of the IraPEN pilot project were included into the study. Sociodemographic characteristics of the population and the prevalence of T2DM are presented

**Table 1. Prevalence of type 2 diabetes by gender, residency and socio-demographic characteristics of the respondents (N = 3691).**

| Variable | Overall n (%) | Diabetes mellitus n (%) | | P |
|---|---|---|---|---|
| **Total prevalence** | | 13.8% | | |
| **Gender** | | | | 0.001 |
| Male | 1625 (44.0%) | 191 (11.8%) | | |
| Female | 2066 (56.0%) | 320 (15.5%) | | |
| **Residency** | | | | 0.073 |
| Rural | 1159 (31.4%) | 143 (12.3%) | | |
| Urban | 2532 (68.6%) | 368 (14.5%) | | |
| **Age** | | Yes | No | <0.001 |
| 30–39 | 807 (21.9%) | 19 (2.4%) | 788 (97.6%) | |
| 40–49 | 1187 (32.2%) | 117 (9.8%) | 1072 (90.2%) | |
| 50–59 | 953 (25.8%) | 203 (21.3%) | 750 (78.7%) | |
| 60+ | 742 (20.1) | 172 (23.2%) | 570 (76.8%) | |
| **Education** | | | | <0.001 |
| Illiterate | 1273 (34.5%) | 264 (20.7%) | 1009 (79.3%) | |
| Elementary | 1000 (27.1%) | 111 (11.1%) | 889 (88.9%) | |
| Middle school | 557 (15.1%) | 62 (11.1%) | 495 (88.9%) | |
| High school | 625 (16.9%) | 55 (8.8%) | 570 (91.2%) | |
| University | 234 (6.3%) | 19 (8.1%) | 215 (91.9%) | |
| **Occupation** | | | | <0.001 |
| Employee | 170 (4.6%) | 10 (5.9%) | 160 (94.1%) | |
| Manual worker | 594 (16.1%) | 66 (11.1%) | 528 (88.9%) | |
| Self-employed | 624 (16.9%) | 67 (10.7%) | 557 (89.3%) | |
| Housewife | 2009 (54.5%) | 319 (15.9%) | 1690 (84.1%) | |
| Unemployed | 208 (5.6%) | 40 (19.2%) | 168 (80.8%) | |
| Other | 82 (2.2%) | 9 (11.0%) | 73 (89.0%) | |
| **Ethnicity** | | | | <0.001 |
| Turk | 1834 (59.9%) | 297 (16.2%) | 1537 (83.8%) | |
| Kurd | 1226 (40.1) | 138 (11.3%) | 1088 (88.7%) | |
| **Marital status** | | | | <0.001 |
| Married | 3319 (89.9%) | 438 (13.2%) | 2881 (86.8%) | |
| Single | 101 (2.7%) | 12 (11.9%) | 89 (88.1%) | |
| Widow | 256 (6.9%) | 59 (23.0%) | 197 (77.0%) | |
| Divorced | 14 (0.4%) | 2 (14.3%) | 12 (85.7%) | |
| **Smoking (yes)** | 321 (8.7%) | 38 (11.8%) | 283 (88.2%) | 0.276 |
| **Alcohol use (yes)** | 15 (0.4%) | 1 (6.7%) | 14 (93.3%) | 0.420 |
| **History of hypertension (yes)** | 596 (16.1%) | 181 (30.4%) | 415 (69.6%) | <0.001 |
| **Family history of diabetes (yes)** | 689 (18.7%) | 168 (24.4%) | 521 (75.6%) | <0.001 |

in Table 1. Thirty-four percent of the population were illiterate. Nearly 90% were married. The highest prevalence of T2DM was among the population over 60 years (23.2%). The prevalence of T2DM was higher among Turks (16.2%) than Kurds (11.3%) (P<0.001) (Table 1).

The overall prevalence of T2DM within the population was 13.8%. Fifty-six percent of the overall population were female. The prevalence of T2DM was higher among women (15.5%) than men (11.8%) (P = 0.001). Approximately 69% of the population lived in urban areas. However, the prevalence of T2DM was higher among urban population (14.5%) than rural population (12.3%), the difference however was not statistically significant (P = 0.073) (Table 1).

**Table 2. The prevalence of risk factors of the type-2 diabetes mellitus by gender and residency.**

| Risk Factors | Gender | | P | Residency | | P | Total |
|---|---|---|---|---|---|---|---|
| | Male | Female | | Rural | Urban | | |
| Smoking | 18.8% | 0.8% | <0.001 | 7.5% | 9.2% | 0.083 | 8.7% |
| Alcohol consumption | 0.8% | 0.1% | 0.001 | 0.1% | 0.6% | 0.039 | 0.4% |
| High blood pressure | 15.2% | 24.0% | <0.001 | 14.2% | 22.8% | <0.001 | 20.1% |
| Overweight | 47.1% | 36.6% | <0.001 | 38.5% | 42.5% | 0.02 | 41.3% |
| Obesity | 23.6% | 50.2% | <0.001 | 36.7% | 39.3% | 0.128 | 38.5% |
| Abdominal obesity | 64.8% | 70.8% | 0.001 | 67.7% | 68.9% | 0.517 | 51.2% |
| High blood cholesterol | 31.8% | 41.9% | <0.001 | 28.5% | 41.6% | <0.001 | 37.4% |
| High triglyceride | 48.0% | 40.4% | <0.001 | 43.7% | 43.8% | 0.975 | 42.5% |

## Prevalence of T2DM risk factors by gender and residency

The prevalence of high blood pressure (P<0.001), obesity (P<0.001), abdominal obesity (P<0.001), and high blood cholesterol (P<0.001) were significantly higher in women. The prevalence of smoking (P<0.001), alcohol consumption (P = 0.001), overweight (P<0.001), and high blood triglycerides (P<0.001) were significantly higher in men. Also, the prevalence of all T2DM-associated risk factors was higher in urban areas, compared to rural areas; alcohol consumption (P = 0.039), high blood pressure (P<0.001), overweight, (P = 0.02) and high blood cholesterol (P<0.001) showed a statistically significant difference. The results showed that the most common risk factor in the population is abdominal obesity (Table 2).

## Determinants of T2DM by gender and residency

The results of simple logistic regression model showed that age, high blood pressure, obesity, abdominal obesity, high blood cholesterol, and triglycerides in both genders had a significant association with the chance of developing T2DM (Table 3). According to the results of multivariate logistic regression, having high blood pressure showed a strong association with the risk of developing T2DM in men (OR 1.77, 95% CI: 1.13–2.79; P = 0.013) and women (OR 2.86, 95% CI: 2.12–3.85; P<0.001). Also, age and having high blood triglycerides showed significant associations with the risk of developing T2DM in both genders. There was a significant relationship between abdominal obesity and the chance of developing T2DM, only among women (OR 1.68, 95% CI: 1.17–2.40; P = 0.004) (Table 3).

The results of simple regression analysis showed that age, high blood pressure, being overweight, abdominal obesity, and high blood cholesterol significantly increased the risk of developing T2DM, in both rural and urban areas (Table 4). Multiple regression analysis showed that, in rural areas, age (OR 1.03, 95% CI: 1.01–1.04; P<0.001), high blood pressure (OR 3.14, 95% CI: 2.0–4.93; P<0.001), abdominal obesity (OR 2.34, 95% CI: 1.41–3.87; P<0.001), and high blood cholesterol (OR 1.59, 95% CI: 1.07–2.37; P = 0.02) had significant associations with the chance of developing T2DM. In urban areas, age (OR 1.02, 95% CI: 1.01–1.04; P<0.001), high blood pressure (OR 2.23, 95% CI: 1.66–3; P<0.001), abdominal obesity (OR 1.46, 95% CI: 06–2.01; P = 0.019) and high triglycerides (OR 1.51, 95% CI: 1.16–1.98; P = 0.002) showed significant relationships with the chance of developing T2DM (Table 4).

**Table 3. The odds ratio of risk factors for developing T2DM by gender.**

| Risk Factors | Status | Male | | Female | | Male | | Female | |
|---|---|---|---|---|---|---|---|---|---|
| | | OR (95% CI) | p-value* | OR (95% CI) | p-value* | OR (95% CI) | p-value** | OR (95% CI) | p-value** |
| Age | Year | 1.03 (1.02–1.04) | <0.001 | 1.06 (1.05–1.07) | <0.001 | 1.01 (1.00–1.03) | 0.012 | 1.03 (1.02–1.04) | <0.001 |
| Residency | Rural | 1 | | 1 | | - | - | - | |
| | Urban | 1.078 (0.77–1.49) | 0.65 | 1.28 (0.98–1.68) | 0.068 | - | - | - | |
| Smoking | No | 1 | | 1 | | - | - | - | |
| | Yes | 0.967 (0.65–1.42) | 0.867 | 1.26 (0.35–4.45) | 0.718 | - | - | - | |
| Alcohol consumption | No | 1 | | 1 | | - | - | - | |
| | Yes | 0 | 0.999 | 5.47 (0.34–87.68) | 0.23 | - | - | - | |
| Blood pressure | Normal | 1 | | 1 | | 1 | | 1 | |
| | High | 2.534 (1.78–3.59) | <0.001 | 4.93 (3.84–6.34) | <0.001 | 1.77 (1.13–2.79) | 0.013 | 2.86 (2.12–3.85) | <0.001 |
| BMI | Underweight and normal | 1 | | 1 | | 1 | | 1 | |
| | Overweight | 1.18 (0.87–1.59) | 0.283 | 0.98 (0.76–1.25) | 0.875 | | | | |
| | Obese | 1.488 (1.06–2.07) | 0.019 | 1.31 (1.03–1.67) | 0.026 | 1.23 (0.82–1.84) | 0.308 | 1.20 (0.90–1.59) | 0.197 |
| Waist-hip ratio | Normal | 1 | | 1 | | 1 | | 1 | |
| | Abdominal obesity | 1.70 (1.1–2.5) | 0.009 | 2.51 (1.79–3.52) | <0.001 | 1.49 (0.99–2.25) | 0.056 | 1.68 (1.17–2.40) | 0.004 |
| Total Cholesterol | Normal | 1 | | 1 | | 1 | | 1 | |
| | High | 1.47 (1.07–2.0) | 0.016 | 1.29 (1.02–1.65) | 0.034 | 1.20 (0.82–1.77) | 0.343 | 0.98 (0.74–1.29) | 0.904 |
| Triglycerides | Normal | 1 | | 1 | | 1 | | 1 | |
| | High | 1.49 (1.1–2.03) | 0.01 | 1.67 (1.31–2.13) | <0.001 | 1.46 (1.01–2.11) | 0.04 | 1.34 (1.02–1.77) | 0.035 |

*Simple logistic regression model.

**Multiple logistic regression models.

Stepwise Forward variable selection method for the multivariate logistic regression analysis. The variables with p< .05 in the simple regression analysis were included in the multiple models.

BMI: Body mass index, OR = Odds Ratio, CI = confidence interval.

## Discussion

### Overall prevalence of T2DM by gender and residency

The purpose of this cross-sectional study was to estimate the prevalence of T2DM and its determinants by gender and residency on over 3000 participants in Naghadeh, Iran. Our findings showed that the prevalence of T2DM in the population with 30 years of age and older was 13.8%, which was significantly higher in women than men, and in urban residents than rural residents. In the study conducted by Afkhami et al, the prevalence of T2DM in people older than 30 years was 14.5%, which is consistent with our findings [30]. Based on another study conducted in Yazd, Iran, the prevalence of T2DM in people older than 30 years was 17.2% [31]. In a meta-analysis on Iranian studies published from 1996 to 2004, the overall prevalence of diabetes among people with 40 years of age and older was reported to be 24% [4]. Such discrepancies can be explained by the variety of the studies, in terms of study type, socio-demographic characteristics, target population, and sample size. Differences in cultural components

**Table 4. The odds ratio of risk factors for developing T2DM by residency.**

| Risk Factors | Status | Rural | | Urban | | Rural | | Urban | |
|---|---|---|---|---|---|---|---|---|---|
| | | OR (95% CI) | p-value* | OR (95% CI) | p-value* | OR (95% CI) | p-value** | OR (95% CI) | p-value** |
| Age | Year | 1.04 (1.03–1.05) | <0.001 | 1.05 (1.04–1.06) | <0.001 | 1.03 (1.01–1.04) | <0.001 | 1.02 (1.01–1.04) | <0.001 |
| Gender | Male | 1 | | 1 | | 1 | | 1 | |
| | Female | 1.21 (0.84–1.72) | 0.292 | 1.44 (1.14–1.81) | 0.002 | 0 (0–0) | 0 | 1.22 (0.90–1.65) | 0.185 |
| Smoking | No | 1 | | 1 | | - | - | - | - |
| | Yes | 1.27 (0.68–2.36) | 0.443 | 0.68 (0.44–1.04) | 0.081 | - | - | - | - |
| Alcohol consumption | No | 1 | | 1 | | - | - | - | - |
| | Yes | 0 (0–0) | 1 | 0.45 (0.05–3.45) | 0.443 | - | - | - | - |
| Blood pressure | Normal | 1 | | 1 | | 1 | - | 1 | - |
| | High | 4.56 (3.08–6.76) | <0.001 | 3.8 (3.0–4.79) | <0.001 | 3.14 (2.0–4.93) | <0.001 | 2.23 (1.66–3) | <0.001 |
| BMI | Underweight and normal | 1 | | 1 | | 1 | | 1 | |
| | Overweight | 1.47 (1.03–2.09) | 0.033 | 1.44 (1.16–1.80) | 0.001 | | | | |
| | Obese | - | - | - | - | 1.32 (0.89–1.96) | 0.167 | 1.12 (0.84–1.48) | 0.422 |
| Waist-hip ratio | Normal | 1 | | 1 | | 1 | | 1 | |
| | Abdominal obesity | 2.95 (1.82–4.79) | <0.001 | 1.91 (1.41–2.59) | <0.001 | 2.34 (1.41–3.87) | 0.001 | 1.46 (1.06–2.01) | 0.019 |
| Total Cholesterol | Normal | 1 | | 1 | | 1 | | 1 | |
| | High | 1.72 (1.19–2.49) | 0.003 | 1.26 (1.0–1.57) | 0.042 | 1.59 (1.07–2.37) | 0.02 | 0.86 (0.65–1.13) | 0.293 |
| Triglycerides | Normal | 1 | | 1 | | 1 | | 1 | |
| | High | 1.40 (0.98–1.99) | 0.061 | 1.63 (1.30–2.03) | <0.001 | 0 (0–0) | 0 | 1.51 (1.16–1.98) | 0.002 |

*Simple logistic regression model.

**Multiple logistic regression models.

Stepwise Forward variable selection method for the multivariate logistic regression analysis. The variables with p< .05 in the simple regression analysis were included in the multiple models.

BMI: Body mass index, OR = Odds Ratio, CI = confidence interval.

of communities may lead to the formation of different behavioral patterns, which might affect their health by shaping different beliefs and attitudes [32]. Due to the lack of previous prevalence studies in Naghadeh, we could not evaluate the trend of T2DM prevalence over time in the population.

Naghadeh is composed of two main ethnicities, including Turks and Kurds. Our results showed that the prevalence of T2DM significantly differed between Kurds (11.3%) and Turks (16.2%), which can be attributed, in part, to the greater number of Turkish participants (59.1%). As the second largest ethnic group in Iran, the most of Iranian Turk and Kurd populations reside in the northwest of Iran (the geographical location of the present study). In a previous study, Abbasi et al. found significant discrepancies among seven ethnic groups in Iran, in terms of the modifiable risk factors, and the severity of Coronary Artery Disease (CAD). In their study, Turk ethnic group were found with higher prevalence of T2DM and severity of CAD, compared to Kurd ethnic group [33]. These findings were similar to those found in our study. In another study that Rezazadeh et al. conducted in northwestern of Iran (Urmia, a city close to Naghadeh), a significant difference was found in dietary patterns between Turk and Kurd participants [34]. As they reported, traditional high socio-economic status patterns and traditional low socio-economic status dietary patterns were more common among Turk and Kurd respondents, respectively, which may be considered as a reason for the differences found in the present study.

Greater prevalence of T2DM in women has also been reported in previous studies conducted in Iran [4,31,35]. Based on the study of Esteghamati et al., the prevalence of diabetes

was higher in women than men [35]. Some of these gender differences may be attributed to sex hormones [13]. Moreover, obesity thresholds in women and insulin resistance in men can explain some of these differences in T2DM prevalence [36]. As in the present study, the prevalence of obesity and abdominal obesity was higher among women compared to men. In our study, there was a significant difference in all T2DM related risk factors by gender (Table 2) that can also explain the difference in the prevalence of T2DM between genders.

Many previous studies have shown differences in the prevalence of T2DM between urban and rural areas [15–17,35]. In agreement with the results of our study, Esteghamati et al. reported that the prevalence of T2DM was higher in urban areas than in rural areas [35]. Urbanization seems to have contributed to an increase in diabetes-related risk factors by increasing some unhealthy behaviors and Western lifestyles [15–17]. In the present study, all diabetes-related risk factors were more prevalent among urban population compared to the rural population. This may explain the relatively higher prevalence in urban areas than in rural areas. Despite the difference in the prevalence of T2DM in urban (14.5%) and rural (12.3%) areas in the present study, this difference was not statistically significant. It may be due to changes in rural lifestyles and their gradual shift to urban lifestyles.

## Determinants of T2DM by gender

Our results showed age, high blood pressure, and high blood triglycerides in associations with the chance of developing T2DMin both men and women. Age and gender are unchangeable risk factors for diabetes [7]. The risk of developing T2DM increases with aging in both genders [4]. In a study performed by Navipur et al., the odds of progression of T2DM increased by aging (OR = 1.28) [37], which is consistent with those found in our study. The results of another study showed that older adults are at high risk for developing T2DM [24]. Due to the effects of increasing insulin resistance and impaired pancreatic function with aging, older people are at high risk for the development of T2DM [38]. It seems that with increase in older adult population, an increase in the prevalence of T2DM is expected. In the present study, the frequency of smoking was 18.8% and 0.8% among men and women, respectively. In a previous study [39], the prevalence of smoking in the northwestern area of Iran (Naghadeh is located in this area) was reported to be less than 1% in women, and from 15 to 30% in men, which is comparable to those found in our study. However, due to the self-report nature of the data, report bias is cautioned for the factors like smoking and alcohol consumption.

High blood pressure increased the chance of developing T2DM among both male and female respondents. Although in our study the prevalence of hypertension was significantly higher in women, the disease was a strong predictor of diabetes in both genders. Meysamie et al., also, reported the prevalence of hypertension to be higher in women than men [35]. In more than two-thirds of the patients with T2DM, hypertension was also reported, and its development coincided with hyperglycemia. Insulin resistance is a disorder that is common patients with high blood pressure and T2DM [40]. It may be a possible reason for the coexistence of hypertension and T2DM. Hypertriglyceridemia is previously reported as a common problem among T2DM patients [41]. Blood triglyceride level is also an independent risk factor and predictor for T2DM [42]. Although high level of blood triglyceride was a determinant of T2DM in both genders in our study, the results showed that the prevalence of high blood triglyceride level was significantly higher in men than women (p <0.001). A study on adolescents showed that the high levels of triglyceride were more common in boys than girls [43]. Differences in blood triglyceride levels can be attributed to different eating patterns between genders. Unhealthy eating habits are more common among men. Men tend to eat more high-fat, high-protein foods than women [44]. Despite the higher prevalence of hypertriglyceridemia among

men [44,45], obesity and abdominal obesity were more common among women in our study, which may be attributed to the higher inactivity of female participants. Since a majority of women in Naghadeh were housewives, we expected that inactivity to be more prevalent among women than men. Thus, obesity and abdominal obesity were higher in women.

### Determinants of T2DM by residency

In the present study, age, high blood pressure, and waist-to-hip ratio (abdominal obesity) in both rural and urban areas, and high levels of blood cholesterol in rural areas, and high triglyceride levels in urban areas, were significant predictors of T2DM. Similar to our findings, high blood pressure is reported to increase the chance of developing T2DM in both rural and urban areas [46]. Over time, T2DM can damage blood vessels, causing their walls to be stiffen, which leads to an increase in blood pressure [47]. In the present study, hypertension was also more prevalent among urban than rural residents, which was similar to those reported by Meysamie et al [17]. Urbanization is noted as a major social determinant of hypertension [48]. Due to the higher prevalence of hypertension risk factors (such as obesity, smoking, high levels of blood cholesterol, and triglycerides) in urban areas, the higher level of blood pressure in urban populations, compared to rural populations, is predictable. Moreover, lifestyle changes [17], and high levels of stress and anxiety due to urbanization [49], can also be attributed to the higher prevalence of hypertension in urban than rural areas.

In the present study, abdominal obesity was a determinant of T2DM in both urban and rural areas. Alam et al. reported that diabetes prevalence is six times higher among individuals with abdominal obesity compared to individuals with normal weight [50]. In the present study, central (intra-abdominal) obesity was observed in the majority of patients with T2DM. A high level of circulating adipokines secreted by abdominal adipose has an essential role in inflammation and insulin resistance [51]. In our study, compared to rural residents, abdominal obesity was higher in urban residents, although the difference was not statistically significant. As previously mentioned, such a difference may be due to the gradual changes that is being made in rural lifestyles.

According to our findings, high blood triglycerides level was a determinant of T2DM in urban areas. In the study of Esteghamati et al., the prevalence of hypertriglyceridemia was higher in urban areas [35], which is consistent with our results. However, the difference in the prevalence of hypertriglyceridemia in rural and urban areas was not statistically significant. The results also showed that the prevalence of abnormal blood cholesterol was significantly higher in urban residents than in rural residents. Similarly, the results of a systematic review showed that 63% of studies reported higher cholesterol levels in urban areas [52]. As mentioned, this could be due to the unhealthy and sedentary lifestyle of urban residents.

Since this study examined the difference in the prevalence of T2DM in urban and rural areas, the results can be generalized to compare the prevalence of T2DM in urban and rural areas in other geographical regions, especially in low and middle-income countries. On the other hand, gender differences in the prevalence of the chronic diseases can also be considered in the generalizability of the results of our study.

### Limitations and strengths of the study

As we conducted this study based on a secondary data collected by the Iranian Ministry of Health, we did not have complete information about the quality of IraPEN program implementation in Naghadeh health centers. Due to the lack of previous studies in the County, it was not possible for us to evaluate the increasing/decreasing trend of T2DM prevalence over time in Naghadeh. Iran as a multi-ethnic country is composed of six major ethnic groups [33].

Since previous studies have shown that the prevalence of T2DM might be different by ethnicity (due to their cultural and lifestyle differences) [53], the results of our study may be mostly generalizable to the populations with Turk and Kurd ethnicities, in Iran, Azerbaijan Republic, Turkey, Iraq and Syria. Despite our efforts, we could not achieve valid data on the distribution of variables in the overall population. Moreover, the male/female proportion of the participants was not so close to those of the overall population of the County, although it may be due to the fact that women have a higher number of primary care visits compared to men [54]. Despite the recommendation of WHO to use digital sphygmomanometer, mercury sphygmomanometer was used in the IraPEN study, as digital blood pressure measuring device was not available in all rural and urban health centers of the country. So, it could be somewhat extent of variation in blood pressure, which might have some potential effect on the results. Concerning with alcohol consumption, the amount of alcohol drinking is reported to be a key influencer for health [55], and moderate alcohol drinking is associated to lower risk of type 2 diabetes [56]. However, in the IraPEN study, the operational definition for alcohol drinking was "having a history of alcohol consumption in the past three months" which was identified to be vague. So, this limitation should be taken into account while studying the results and data interpretation. As another limitation, it leads to impacts of results, validity and data interpretation. Fruit and vegetable consumption is an important factor for diabetes prevalence. However, this data is missing in our results. As a strength, this study was a large community survey conducted on 3691 participants. Another strength was the evaluation of the most important risk factors involved in the prevalence of T2DM by gender and residency.

## Conclusion

The results of the present study showed that the overall prevalence of T2DM in Naghadeh was 13.8%, which was higher among women than men, and in urban areas than rural areas. The recognition of the prevalence of T2DM and its determinants among populations by gender and residency is a key step in establishing community-based interventions for modifying cardiometabolic risk factors in communities. Given the higher prevalence of T2DM among women, risk reduction strategies at the community level should be more targeted at women. Moreover, the higher prevalence of T2DM risk factors among the urban population is a wake-up call for policymakers to pay more attention to the consequences of unhealthy and sedentary urban lifestyles in communities. Our findings might be helpful for health policymakers and health practitioners in making effective and feasible public policies and designing health promotion programs to prevent T2DM, and reduce its-associated co-morbidities and complications. It is recommended that future actions to be focused on appropriate timely action plans for prevention and control of T2DM from the early years of life, especially planning for effective health education and promotion programs.

## Acknowledgments

Authors are grateful to Tabriz University of Medical Sciences, health center of Naghadeh and the health deputy of Urmia University of Medical Sciences for their supports. The authors claim that there are no conflicts of interests. We also thank people of Naghadeh who participated in this study.

## Author Contributions

**Conceptualization:** Nafiseh Ghassab-Abdollahi, Haidar Nadrian, Shayesteh Shirzadi, Parvin Sarbakhsh, Fatemeh Saadati, Leila Zhianfar.

**Data curation:** Kobra Pishbin, Parvin Sarbakhsh, Mohammad Sanyar Moradi.

**Formal analysis:** Nafiseh Ghassab-Abdollahi, Shayesteh Shirzadi, Mohammad Sanyar Moradi, Pouria Sefidmooye Azar, Leila Zhianfar.

**Methodology:** Haidar Nadrian, Kobra Pishbin, Shayesteh Shirzadi.

**Software:** Kobra Pishbin.

**Writing – original draft:** Nafiseh Ghassab-Abdollahi, Kobra Pishbin, Shayesteh Shirzadi, Parvin Sarbakhsh, Fatemeh Saadati, Pouria Sefidmooye Azar, Leila Zhianfar.

**Writing – review & editing:** Nafiseh Ghassab-Abdollahi, Haidar Nadrian, Fatemeh Saadati, Pouria Sefidmooye Azar, Leila Zhianfar.

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
