## [Decision Letter · Decision Letter 0]

17 Jan 2022

PONE-D-21-19349Gender and Urban–Rural Residency based Differences in the Prevalence of Type-2 Diabetes Mellitus and its Determinants among Iranian Adults: Results of IraPEN SurveyPLOS ONE

Dear Dr. Nadrian,

Thank you for submitting your manuscript to PLOS ONE. After careful consideration, we feel that it has merit but does not fully meet PLOS ONE’s publication criteria as it currently stands. Therefore, we invite you to submit a revised version of the manuscript that addresses the points raised during the review process.

We look forward to receiving your revised manuscript.

Kind regards,

Kannan Navaneetham, PhD

Academic Editor

PLOS ONE

https://journals.plos.org/plosone/s/file?id=ba62/PLOSOne_formatting_sample_title_authors_affiliations.pdf”

2. Please clearly indicate within the Methods section that the current study reported within the manuscript is a secondary analysis of the IraPEN project data.

3. ‘In ethics statement in the manuscript and in the online submission form, please provide additional information about the patient records/samples used in your retrospective study. Specifically, please ensure that you have discussed whether all data/samples were fully anonymized before you accessed them and/or whether the IRB or ethics committee waived the requirement for informed consent. If patients provided informed written consent to have data/samples from their medical records used in research, please include this information.

“This research received no specific grant.”

Reviewers' comments:

Reviewer's Responses to Questions

**Comments to the Author**

1. Is the manuscript technically sound, and do the data support the conclusions?

Reviewer #1: No

Reviewer #2: Partly

2. Has the statistical analysis been performed appropriately and rigorously? 

Reviewer #1: No

Reviewer #2: No

3. Have the authors made all data underlying the findings in their manuscript fully available?

Reviewer #1: No

Reviewer #2: Yes

4. Is the manuscript presented in an intelligible fashion and written in standard English?

Reviewer #1: Yes

Reviewer #2: No

5. Review Comments to the Author

Reviewer #1: Thank you very much for your interesting manuscript! However, I have some suggestions for better scientific study.

Major comments

1. Regarding with methodology, the authors reported that they used the data from IraPEN Program in Naqadesh. The eligible criteria for the participants was set up by the census data. I understand that the authors used the secondary data for this study. You should write it clearly for using the secondary data and the study year (IraPEN program and census data) if so.

2. The study was based on from IraPEN Program in Naqadesh. The IraPEN Program was adopted from WHO PEN for primary care actually. Therefore, you should follow the WHO STEP survey straightly for the data collection concerning anthropometric measurement and biological investigation. Currently the variable using in the study were the same as WHO STEP survey, however, the methods to measurements were not strictly follow to WHO STEP guidelines. Thus, it would make the impacts of data validity.

3. The operational definition for smoking and alcohol consumption should be reported in details. You use another reference for it rather than WHO STEP guidelines. Could you report any special reason to do this please?

4. Concerning with fasting blood glucose investigation, you reported that the blood drawing was made twice consecutively. How did you choose the real FBS level of the participants among these two blood samples? Please describe the any special reason to do this?

5. Regarding with plasma cholesterol levels, is it fasting or non-fasting? It would be effect on FBS level on later analysis plan.

6. For the operational definition of T2DM, the authors defined the it (>7.0mmol/l) without using reference. You should use the WHO definition for diagnosis of DM because you used the data from IRA PEN program. In addition, you estimation may vary if you use the operational definition for diagnosis for DM (≥7.0mmol/l).

7. The same comment for operational definition for hypertension.

8. The authors reported that linear logistic regression in table 4 and 6. What do you mean for linear logistic regression?

9. What is the differences between tab 4 and 5, table 6 and 7?

10. To report the odd ratio of risk factors for developing of DM, which advanced methods was used for regression model? How did you adjust bias, confounding, mediators as you say so multiple logistic regression?

11. What is the generalizability of your study?

12. You used the data from IraPEN Program in Naqadesh only. However, your title refers the prevalence of Type 2 DM and its determinants among Iranian Adults. How did you think the population in Naqadesh cover to represent all Iranian adults?

13. It is better to include how to calculate to have this study population?

Minor comments

1. The weakness of the study is not enough yet. You should add it accordingly.

Reviewer #2: This "Gender and Urban–Rural Residency based Differences in the Prevalence of Type-2 Diabetes Mellitus and its Determinants among Iranian Adults: Results of IraPEN Survey" is a good issue for the epidemiologists and policemakers. But I have some question about this article.

1.The major weakness of this study is the representativeness. The study is using the results of IranPEN survey to explore the "Gender and Urban–Rural Residency based Differences in the Prevalence of Type-2 Diabetes Mellitus and its Determinants among Iranian Adults". Based on the survey, a total 3691 participants of 30-70 y/o were included. There were more females(56.0%) in study. Readers even didn't know about the general status in Naqadeh, ex.: the distribution of overall population by gender, age-group and race in Naqadeh. Besides, I also wonder about the representativeness because this survey was done in 2017.

2.The definition of rural & urban is not described clearly. What's the major economic activities in Naghadeh (agriculture, financial industry or others) ? What's the difference of this city from other cities?

3.The study included two largest ethnic groups (Trukic and Kurds). In discussion, the author talked about this mixed ethnic context can influence the prevalence of T2DM compared to other areas. I would wonder about if this two largest ethnic groups have some special characteristics from others. For example, they got higher cardiovascular diseases, thin appearance or special eating cultures. The author didn't discuss about that.

3.Table 1 shows the baseline characteristic of participants. How about the distribution of variables compared to overall population? For example, the distribution of age-group is similar with overall of Naqadeh or not.

4.Table 3 shows about the prevalence of related risk factor of T2DM by gender. I saw the low percentage of smoking status. Is that reasonable in this city?? is it comparable with the real status of Naqadeh.

5.Table 4 & 5 could be combined together. (the same condition with Table 6 &7.)

I agree with the importance of the epidemiology of non-communicable diseases. We have known the prevalence & related risk factors first and make effective and feasible public policies to reduce the related co-morbidities and complications. However, the representativeness of this study is questioned.

6. PLOS authors have the option to publish the peer review history of their article (what does this mean?). If published, this will include your full peer review and any attached files.

Reviewer #1: No

Reviewer #2: No

---

## [Author Response · Author response to Decision Letter 0]

26 Mar 2022

Reviewer #1: Thank you very much for your interesting manuscript! However, I have some suggestions for better scientific study.

Major comments

1. Regarding with methodology, the authors reported that they used the data from IraPEN Program in Naqadesh. The eligible criteria for the participants was set up by the census data. I understand that the authors used the secondary data for this study. You should write it clearly for using the secondary data and the study year (IraPEN program and census data) if so.

- In method section we mentioned "Using the census data" we meant, the research team included all participants who participated in IraPEN project without any sampling. In fact, 3691 people participated in IraPEN project and we included the information of all of them in secondary analysis (similar to a census).

- To reduce any ambiguity, we revised and rewrote the sentences clearly.

- Revisions: Page 7, Lines 180-182 and 188-190.

2. The study was based on from IraPEN Program in Naqadesh. The IraPEN Program was adopted from WHO PEN for primary care actually. Therefore, you should follow the WHO STEP survey straightly for the data collection concerning anthropometric measurement and biological investigation. Currently the variable using in the study were the same as WHO STEP survey, however, the methods to measurements were not strictly follow to WHO STEP guidelines. Thus, it would make the impacts of data validity.

- IraPEN was piloted in 4 main districts of Iran. (Baft, Naghadeh, Shahrreza and Maragheh). 

- The implementation of this program in the health centers of Naghadeh was carried out according to the "Package of essential Noncommunicable diseases in Iran's primary health care system "Irapen"", which was developed based on the WHO PEN program under the supervision of the representative office of the WHO in Iran. This instruction was provided in Persian and all health care centers in Naghadeh followed it. In fact, IraPEN program was a modified version of the STEP wise Approach to NCD Risk Factor Surveillance (STEPS) (1). In other words, the project was adjusted for Iranian setting, and that is why some differences may be seen in the measurements. 

- We rewrote and added the proper reference to method section.

- Revisions: 

- Pages 6 and 7: Lines 163-169; Page 8: Lines 192-194.

3. The operational definition for smoking and alcohol consumption should be reported in details. You use another reference for it rather than WHO STEP guidelines. Could you report any special reason to do this please?

As mentioned above, all steps of IraPEN project were conducted in Naghadeh according to the "Package of essential Noncommunicable diseases in Iran's primary health care system "Irapen"", prepared by the Ministry of Health. We also noted that we worked on a secondary data. However, after contacting the administrative team of the study in Naghadeh, they noted that their data collection process and the instruments used for data collection were all according to the abovementioned package and the directive of the ministry. In IraPEN guideline, the practical definition of smoking and alcohol consumption was having a history of smoking and/or alcohol consumption in the past three months. So, in our present data analysis, this definition was used. 

- The Package of essential Noncommunicable diseases in Iran's primary health care system "Irapen", is provided in Persian, we thought this reference could be unfamiliar to readers, so referred to a (WHO ASSIST Working Group. The Alcohol, Smoking and Substance Involvement Screening Test (ASSIST): Development, reliability and feasibility) (2). 

- Please see: Pages 8- 9, Lines 214-220.

4. Concerning with fasting blood glucose investigation, you reported that the blood drawing was made twice consecutively. How did you choose the real FBS level of the participants among these two blood samples? Please describe the any special reason to do this?

As mentioned in the method section, this study was a secondary data analysis. To answer this comment, we contacted the staff of IraPEN project. They reported that only people with fasting blood glucose above 126 mg / dL were tested twice, if their fasting blood glucose in second measurement was again, 126 mg / dL or more they were diagnosed as a diabetic patient.

Revision

Please see: Page 9, Lines 228-231.

5. Regarding with plasma cholesterol levels, is it fasting or non-fasting? It would be effect on FBS level on later analysis plan.

According to the Package of essential Noncommunicable diseases in Iran's primary health care system "Irapen"(1), prepared by the Ministry of Health this test can be performed fasting or non-fasting. Moreover, lipid profiles at most change minimally in response to normal food intake in individuals in the general population (3).

We added proper sentence to method section. 

Please see: Page 9, Line 235.

6. For the operational definition of T2DM, the authors defined the it (>7.0mmol/l) without using reference. You should use the WHO definition for diagnosis of DM because you used the data from IRA PEN program. In addition, you estimation may vary if you use the operational definition for diagnosis for DM (≥7.0mmol/l).

According to the Package of essential Noncommunicable diseases in Iran's primary health care system "Irapen" the definition of T2DM is Fasting plasma glucose ≥7.0mmol/l (126mg/dl). (1) Moreover, Aaccording to "Implementation tools Package of Essential Noncommunicable (PEN) disease interventions for primary health care in low-resource settings" the current WHO recommendations for the diagnostic criteria for diabetes and intermediate hyperglycaemia is Fasting plasma glucose ≥7.0mmol/l (126mg/dl) (4). 

We provided proper reference in the main text based on WHO guideline for PEN project.

We added reference. Thanks. 

 Please see: Page 9, Lines 226-228.

7. The same comment for operational definition for hypertension.

According to Implementation tools Package of Essential Noncommunicable (PEN) disease interventions for primary health care in low-resource settings the current WHO recommendations for the diagnostic criteria of high blood pressure is BP ≥ 140/90 mm Hg in twice measurements. The average of two measurements was calculated.

We provided proper reference.

 Please see: Page 8, Lines 212-214.

8. The authors reported that linear logistic regression in table 4 and 6. What do you mean for linear logistic regression?

We meant Simple logistic regression not linear. It was written by mistake.

 We corrected it.

Please see: Page 10, Line 241.

9. What is the differences between tab 4 and 5, table 6 and 7?

o As you know, Simple logistic regression (Table 4 and 6) applies when there is a single dichotomous outcome and one independent variable while multiple logistic regression (Table 5 and 7) applies when there is a single dichotomous outcome and more than one independent variable. In fact, we used simple logistic regression to show a crude effect, and multiple logistic regression to show an adjusted effect. In other word, presenting both of these analyzes gives the reader a better insight about variables. Moreover, simple logistic regression helps to find the non-significant variables and entering them to multiple logistic regression. 

o We added more details to Statistical analysis section. 

o According to reviewer 2 comments we combined Table 4 and Table 5;Table 6 and Table 7. 

o Please see: Page 10, Line 242-247.

10. To report the odd ratio of risk factors for developing of DM, which advanced methods was used for regression model? How did you adjust bias, confounding, mediators as you say so multiple logistic regression?

We used Multiple Forward Stepwise Logistic Regression model. In this technique, as you know, we test for the associations of potential confounders to the outcome.

The details added to Statistical analysis section. 

Please see: Page 10, Line 242-247; Tables 4 and 5 legends. 

11. What is the generalizability of your study?

Iran as a multi-ethnic country is composed of six major ethnic groups (5). Since previous studies have shown that the prevalence of T2DM can vary significantly depending on the ethnicity and race of individuals due to their cultural and lifestyle differences (6), so the results of this study may be mostly generalizable to the populations with Turk and Kurd ethnicities. However, it should be noted that Turkish and Kurdish ethnicities are the second and third largest ethnicities in Iran, respectively.

We added it to limitation section.

Please see: Pages 25 and 26, Lines 456-460 and 464-477.

12. You used the data from IraPEN Program in Naqadesh only. However, your title refers the prevalence of Type 2 DM and its determinants among Iranian Adults. How did you think the population in Naqadesh cover to represent all Iranian adults?

Thanks for your comment. We rewrote the title. 

13. It is better to include how to calculate to have this study population?

As mentioned in the method section, this study was a secondary data analysis. So, it is not clear for us how sample size is calculated. To answer this comment, we contacted the staff of IraPEN project. They reported that no formula and/or limit is considered for sample size. Instead, based on the national guideline of the project, they planned to continue the inclusion of population into the project until the full coverage of the County population (census). When we requested the project’s data for our study, they provided us with the data of the first 3691 participants, only.

Minor comments

1. The weakness of the study is not enough yet. You should add it accordingly.

We added further weaknesses. 

Please see: Page 26, Lines 464-477.

Reviewer #2: This "Gender and Urban–Rural Residency based Differences in the Prevalence of Type-2 Diabetes Mellitus and its Determinants among Iranian Adults: Results of IraPEN Survey" is a good issue for the epidemiologists and policemakers. But I have some question about this article.

1.The major weakness of this study is the representativeness. The study is using the results of IranPEN survey to explore the "Gender and Urban–Rural Residency based Differences in the Prevalence of Type-2 Diabetes Mellitus and its Determinants among Iranian Adults". Based on the survey, a total 3691 participants of 30-70 y/o were included. There were more females(56.0%) in study. Readers even didn't know about the general status in Naqadeh, ex.: the distribution of overall population by gender, age-group and race in Naqadeh. Besides, I also wonder about the representativeness because this survey was done in 2017.

We added enough information about general status in Naghadeh. We only have information about the general status in Naqadeh regarding gender. More women than men maybe due to this fact that women have a higher number of primary care visits compared to men (7). 

Please see: Page 7, Lines 172-177; And Page 26 (limitations); lines 464-477.

2.The definition of rural & urban is not described clearly. What's the major economic activities in Naghadeh (agriculture, financial industry or others) ? What's the difference of this city from other A We provided definition of rural & urban in the method section.

Please see: Page 7, Lines 185-188.

3.The study included two largest ethnic groups (Trukic and Kurds). In discussion, the author talked about this mixed ethnic context can influence the prevalence of T2DM compared to other areas. I would wonder about if this two largest ethnic groups have some special characteristics from others. For example, they got higher cardiovascular diseases, thin appearance or special eating cultures. The author didn't discuss about that. 

Please see: Pages 21-22, Lines 354-367.

3.Table 1 shows the baseline characteristic of participants. How about the distribution of variables compared to overall population? For example, the distribution of age-group is similar with overall of Naqadeh or not.

Despite our efforts, we could not achieve a valid data on the distribution of variables in the overall population

.We added it as a limitation to the manuscript. 

Please see: Page 26 (limitations); lines 464-477.

4.Table 3 shows about the prevalence of related risk factor of T2DM by gender. I saw the low percentage of smoking status. Is that reasonable in this city?? is it comparable with the real status of Naqadeh.

In a previous study (8), the prevalence of smoking in 2016 in the northwestern provinces of Iran (Naghadeh is located in the northwestern of Iran ) was reported less than one percent in women and between 15-30 percent in men which is somewhat in accordance with our findings. 

Please see: Page 23, lines 396-401.

5.Table 4 & 5 could be combined together. (the same condition with Table 6 &7.)

We combined them together.

Please see: Tables 4 and 5, Pages 15-16 AND 18-19.

I agree with the importance of the epidemiology of non-communicable diseases. We have known the prevalence & related risk factors first and make effective and feasible public policies to reduce the related co-morbidities and complications. However, the representativeness of this study is questioned.

Thanks for the comment.

Iran as a multi-ethnic country is composed of six major ethnic groups (5). Since previous studies have shown that the prevalence of T2DM can vary significantly depending on the ethnicity and race of individuals due to their cultural and lifestyle differences (6), so the results of this study may be generalizable only in two Iranian ethnicities. However, it should be noted that Turkish and Kurdish ethnicities are the second and third largest ethnicities in Iran, respectively.

We also added the representativeness issue of the study as a limitation to the limitation section.

Please see: Pages 25 and 26, Lines 456-460 and 464-477.

6. PLOS authors have the option to publish the peer review history of their article (what does this mean?). If published, this will include your full peer review and any attached files.

Do you want your identity to be public for this peer review? For information about this choice, including consent withdrawal, please see our Privacy Policy.

Reviewer #1: No

Reviewer #2: No

References

1. A K, M N, A MH, A M, Ghanbari, A M, et al. Package of essential Noncommunicable diseases in Iran’s primary health care system “Irapen.” 1 st ed. Tehran: Mojasameh; 2017. 

2. Group WAW. The Alcohol, Smoking and Substance Involvement Screening Test (ASSIST): Development, reliability and feasibility. Addiction. 2002;97(9):1183–94. 

3. Langsted A, Freiberg JJ, Nordestgaard BG. Fasting and Nonfasting Lipid Levels. Circulation. 2008;118(20):2047–56. 

4. World Health Organisation. Implementation tools Package of Essential Noncommunicable (PEN) disease interventions for primary health care in low-resource settings. MscforumOrg. 2013;1–210. 

5. Abbasi SH, Sundin Ö, Jalali A, Soares J, Macassa G. Ethnic Differences in the Risk Factors and Severity of Coronary Artery Disease: a Patient-Based Study in Iran. Journal of Racial and Ethnic Health Disparities. 2018;5(3):623–31. 

6. Cheng YJ, Kanaya AM, Araneta MRG, Saydah SH, Kahn HS, Gregg EW, et al. Prevalence of Diabetes by Race and Ethnicity in the United States, 2011-2016. JAMA - Journal of the American Medical Association. 2019;322(24):2389–98. 

7. Bertakis KD, Azari R, Helms LJ, Callahan EJ RJ. Gender differences in the utilization of health care services. J Fam Pract. 2000;49(2):147–52. 

8. Sohrabi MR, Abbasi-Kangevari M, Kolahi AA. Current Tobacco Smoking Prevalence Among Iranian Population: A Closer Look at the STEPS Surveys. Frontiers in Public Health. 2020;8.

---

## [Decision Letter · Decision Letter 1]

25 Apr 2022

PONE-D-21-19349R1Gender and Urban–Rural Residency based Differences in the Prevalence of Type-2 Diabetes Mellitus and its Determinants among Adults in Naghadeh : Results of IraPEN SurveyPLOS ONE

Dear Dr. Nadrian,

Thank you for submitting your manuscript to PLOS ONE. After careful consideration, we feel that it has merit but does not fully meet PLOS ONE’s publication criteria as it currently stands. Therefore, we invite you to submit a revised version of the manuscript that addresses the points raised during the review process.

We look forward to receiving your revised manuscript.

Kind regards,

Kannan Navaneetham, PhD

Academic Editor

PLOS ONE

Journal Requirements:

Reviewers' comments:

Reviewer's Responses to Questions

**Comments to the Author**

1. If the authors have adequately addressed your comments raised in a previous round of review and you feel that this manuscript is now acceptable for publication, you may indicate that here to bypass the “Comments to the Author” section, enter your conflict of interest statement in the “Confidential to Editor” section, and submit your "Accept" recommendation.

Reviewer #1: All comments have been addressed

Reviewer #2: All comments have been addressed

2. Is the manuscript technically sound, and do the data support the conclusions?

Reviewer #1: Partly

Reviewer #2: Yes

3. Has the statistical analysis been performed appropriately and rigorously? 

Reviewer #1: No

Reviewer #2: Yes

4. Have the authors made all data underlying the findings in their manuscript fully available?

Reviewer #1: Yes

Reviewer #2: Yes

5. Is the manuscript presented in an intelligible fashion and written in standard English?

Reviewer #1: Yes

Reviewer #2: Yes

6. Review Comments to the Author

Reviewer #1: Thank you for your interesting manuscript according to the comments! However, I have some suggestions for it.

1. Table 1 and 2 should be combined and presented it as in one table.

2. I understand that the table 3 showed the prevalence of risk factors of the diabetes (DM) via gender and residency. However, the meaning you wrote is a little bit different if my thinking is right. Thus, be updated please.

3. The weakness concerning the methodology of the study must be reported under the title of the limitation and strength of the study.

Reviewer #2: The author has addressed all my comments.But I still have some suggestions.

1.About the revision table 4 & 5, stepwise forward variable selection method was done. But for clinical epidemiology, the important variables should been preserved even if they didn't get significant p value, ex. smoking, alcohol...etc.

2.Some error wording should be corrected. For example, the "exercised" on p25 line 473. That should be "concerned" or anything else?! I advise to send this manuscript to the professional English editing.

7. PLOS authors have the option to publish the peer review history of their article (what does this mean?). If published, this will include your full peer review and any attached files.

Reviewer #1: No

Reviewer #2: No

---

## [Author Response · Author response to Decision Letter 1]

12 Jul 2022

Comments to the Author

1. If the authors have adequately addressed your comments raised in a previous round of review and you feel that this manuscript is now acceptable for publication, you may indicate that here to bypass the “Comments to the Author” section, enter your conflict of interest statement in the “Confidential to Editor” section, and submit your "Accept" recommendation.

Reviewer #1: All comments have been addressed

Reviewer #2: All comments have been addressed

2. Is the manuscript technically sound, and do the data support the conclusions?

Reviewer #1: Partly

Reviewer #2: Yes

3. Has the statistical analysis been performed appropriately and rigorously?

Reviewer #1: No

Reviewer #2: Yes

4. Have the authors made all data underlying the findings in their manuscript fully available?

Reviewer #1: Yes

Reviewer #2: Yes

5. Is the manuscript presented in an intelligible fashion and written in standard English?

Reviewer #1: Yes

Reviewer #2: Yes

6. Review Comments to the Author

Reviewer #1: Thank you for your interesting manuscript according to the comments! However, I have some suggestions for it.

1. Table 1 and 2 should be combined and presented it as in one table.

 -We merged them. 

 Please see: Page 11, Table 1.

2. I understand that the table 3 showed the prevalence of risk factors of the diabetes (DM) via gender and residency. However, the meaning you wrote is a little bit different if my thinking is right. Thus, be updated please.

-We rewrote it.

Please see: Page 13.

3. The weakness concerning the methodology of the study must be reported under the title of the limitation and strength of the study.

We reported it under the title of the limitation and strength of the study. 

Please see: Page 25, Lines 457-462.

Reviewer #2: The author has addressed all my comments. But I still have some suggestions.

1.About the revision table 4 & 5, stepwise forward variable selection method was done. But for clinical epidemiology, the important variables should been preserved even if they didn't get significant p value, ex. smoking, alcohol...etc.

Thanks for the comment. As we performed stepwise forward regression, the insignificant variables were omitted in the next step of analysis with the hope to provide the readers with a clear presentation of the ORs. Moreover, the insignificant role of such important variables is presented in the first section of the table, where the results of the first step of regression is reported. So, the reader can get information from those variables, as well. 

2.Some error wording should be corrected. For example, the "exercised" on p25 line 473. That should be "concerned" or anything else?! I advise to send this manuscript to the professional English editing.

It was corrected.

Please see: Page 25, Line 461.

The manuscript was edited by a professional English editor.

7. PLOS authors have the option to publish the peer review history of their article (what does this mean?). If published, this will include your full peer review and any attached files.

Do you want your identity to be public for this peer review? For information about this choice, including consent withdrawal, please see our Privacy Policy.

Reviewer #1: No

Reviewer #2: No

---

## [Decision Letter · Decision Letter 2]

20 Sep 2022

PONE-D-21-19349R2Gender and Urban–Rural Residency based Differences in the Prevalence of Type-2 Diabetes Mellitus and its Determinants among Adults in Naghadeh: Results of IraPEN SurveyPLOS ONE

Dear Dr. Nadrian,

Thank you for submitting your manuscript to PLOS ONE. After careful consideration, we feel that it has merit but does not fully meet PLOS ONE’s publication criteria as it currently stands. Therefore, we invite you to submit a revised version of the manuscript that addresses the points raised during the review process. Reviewer one still has a number of concerns about the methodology in particular which should be fully addressed.

We look forward to receiving your revised manuscript.

Kind regards,

Alice Coles-Aldridge

Staff Editor

PLOS ONE

Reviewers' comments:

Reviewer's Responses to Questions

**Comments to the Author**

1. If the authors have adequately addressed your comments raised in a previous round of review and you feel that this manuscript is now acceptable for publication, you may indicate that here to bypass the “Comments to the Author” section, enter your conflict of interest statement in the “Confidential to Editor” section, and submit your "Accept" recommendation.

Reviewer #1: (No Response)

Reviewer #2: All comments have been addressed

2. Is the manuscript technically sound, and do the data support the conclusions?

Reviewer #1: Partly

Reviewer #2: Yes

3. Has the statistical analysis been performed appropriately and rigorously? 

Reviewer #1: No

Reviewer #2: Yes

4. Have the authors made all data underlying the findings in their manuscript fully available?

Reviewer #1: No

Reviewer #2: Yes

5. Is the manuscript presented in an intelligible fashion and written in standard English?

Reviewer #1: Yes

Reviewer #2: Yes

6. Review Comments to the Author

Reviewer #1: Thank you very much for your interesting manuscript! However, I still have some suggestions for clarification.

Major comments concerning methodology

1. The mercury sphygmomanometer was reported to be used for the assessment of blood pressure of the participants. This means the deviation from the specific blood pressure while not using the digital sphygmomanometer according to WHO. Thus, how did you control it concerning with this result?

2. Regarding with the operational definition of the variables, the authors reported to use reference number 28. Actually, other WHO references (for example, the following references) must be used rather than reference no 28 because there is no specific definitions for your variables.

For hypertension : World Health Organization. WHO STEPS surveillance manual: the WHO STEPwise approach to chronic disease risk factor surveillance. 2008. http://www.who.int/chp/steps/en/.

For diabetes: World Health Organization. Definition and diagnosis of diabetes mellitus and and intermediate hyperglycemia, 2006.

You can see the much differences of your variables’ definitions while comparing with above references although you said you cited the reference as Package of Essential

Noncommunicable (PEN) diseases interventions for primary health care in low-resource settings. It would be the effect for the validity of your results.

3. In addition, you mentioned the condition that you might have repeated fasting blood glucose for some participants. However, you did not describe when you take the blood again for fasting blood glucose (immediately or next day morning). This also effect to your results. Concerning with lipid profiles, you reported that the results are mixed with fasting and non-fasting. This means the results of random lipid profile.

4. The random lipid profiles are difficult to be use and interpret rather than the result of fasting lipid profile for the scientific paper. The results concerning lipid profile was random results according to your report.

5. Concerning with alcohol drinking, the amount of alcohol drinking is one of key influencer for the health. Moderate alcohol drinking is associated lower risk of type 2 diabetes, revealed in many research papers. Therefore, your operational definition for alcohol is vague, it leads to impacts of results, validity and data interpretation.

6. It is enough when prevalence of diabetes is described in the table 1.

7. You reported the odds ratio for age in the table 4 and 5. As you mentioned you used multiple logistic regression for both table. However, I understand that you used the independent variable äge” as the continuous variable for the analysis according to your description in the table. If so, it cannot be presented as odds ratio.

8. You reported that you applied the stepwise forward approach for logistic regression in the table 4 and 5. You should compare to present with crude model. In addition, you also could check the best fit models?

Reviewer #2: All comments have been addressed. I got no more comments for the authors. They have also finish the English editing.

7. PLOS authors have the option to publish the peer review history of their article (what does this mean?). If published, this will include your full peer review and any attached files.

Reviewer #1: No

Reviewer #2: No

---

## [Author Response · Author response to Decision Letter 2]

10 Nov 2022

Comments to the Author

1. If the authors have adequately addressed your comments raised in a previous round of review and you feel that this manuscript is now acceptable for publication, you may indicate that here to bypass the “Comments to the Author” section, enter your conflict of interest statement in the “Confidential to Editor” section, and submit your "Accept" recommendation.

Reviewer #1: (No Response)

Reviewer #2: All comments have been addressed

2. Is the manuscript technically sound, and do the data support the conclusions?

Reviewer #1: Partly

Reviewer #2: Yes

3. Has the statistical analysis been performed appropriately and rigorously?

Reviewer #1: No

Reviewer #2: Yes

4. Have the authors made all data underlying the findings in their manuscript fully available?

Revisions: 

Our data is fully available. We uploaded our data on the Figshare site to be publicly available.

Reviewer #1: No

Reviewer #2: Yes

5. Is the manuscript presented in an intelligible fashion and written in standard English?

Reviewer #1: Yes

Reviewer #2: Yes

6. Review Comments to the Author

Reviewer #1: Thank you very much for your interesting manuscript! However, I still have some suggestions for clarification.

Major comments concerning methodology

1. The mercury sphygmomanometer was reported to be used for the assessment of blood pressure of the participants. This means the deviation from the specific blood pressure while not using the digital sphygmomanometer according to WHO. Thus, how did you control it concerning with this result?

Revisions: : 

Materials and methods; Data collection and measurements; paragraph 3; lines 2-5.

.Regarding with the operational definition of the variables, the authors reported to use reference number 28. Actually, other WHO references (for example, the following references) must be used rather than reference no 28 because there is no specific definitions for your variables.

For hypertension : World Health Organization. WHO STEPS surveillance manual: the WHO STEPwise approach to chronic disease risk factor surveillance. 2008. http://www.who.int/chp/steps/en/.

For diabetes: World Health Organization. Definition and diagnosis of diabetes mellitus and and intermediate hyperglycemia, 2006.

You can see the much differences of your variables’ definitions while comparing with above references although you said you cited the reference as Package of Essential Noncommunicable (PEN) diseases interventions for primary health care in low-resource settings. It would be the effect for the validity of your results.

Revisions: 

Thank you for your comment. The appropriate references were replaced.

In addition, you mentioned the condition that you might have repeated fasting blood glucose for some participants. However, you did not describe when you take the blood again for fasting blood glucose (immediately or next day morning). This also effect to your results. Concerning with lipid profiles, you reported that the results are mixed with fasting and non-fasting. This means the results of random lipid profile.

Revisions: 

Materials and methods; Data collection and measurements; paragraph 5; lines 4-5.

Thanks for the comment, the lipid profile test was also taken in a fasting condition.

We rewrote the sentence and removed ambiguous sentence.

4. The random lipid profiles are difficult to be use and interpret rather than the result of fasting lipid profile for the scientific paper. The results concerning lipid profile was random results according to your report.

Revisions: 

Materials and methods; Data collection and measurements; paragraph 5; lines 6-7.

Blood sugar and lipid profile tests were taken in a same visit and all participants were fasting for the blood sugar test, so, the lipid profile test was also taken in a fasting condition.

. Concerning with alcohol drinking, the amount of alcohol drinking is one of key influencer for the health. Moderate alcohol drinking is associated lower risk of type 2 diabetes, revealed in many research papers. Therefore, your operational definition for alcohol is vague, it leads to impacts of results, validity and data interpretation.

Revisions: 

As we mentioned before, the current study was a secondary data analysis. In IraPEN guideline, the practical definition of alcohol consumption was having a history of alcohol consumption in the past three months. So, in our present data analysis, these data were analyzed. However, we noted your comment as a limitation in the limitation section.

Discussion; limitations and strengths; paragraph 1; lines 12-17.

It is enough when prevalence of diabetes is described in the table 1.

Yes, Diabetes prevalence is reported in Table 1.

. You reported the odds ratio for age in the table 4 and 5. As you mentioned you used multiple logistic regression for both table. However, I understand that you used the independent variable äge” as the continuous variable for the analysis according to your description in the table. If so, it cannot be presented as odds ratio.

Yes, you are right. But, Logistic regression, like ordinary regression, can have multiple explanatory variables, as Agresti reported (1). Some or all of those predictors can be categorical, rather than quantitative. An important interpretation of the logistic regression model uses the odds ratio [1]. So, we can use odds ratio as one of the ways to interpret such logistic regression analyses. 

. You reported that you applied the stepwise forward approach for logistic regression in the table 4 and 5. You should compare to present with crude model. In addition, you also could check the best fit models?

Thanks for your great comment. Since the aim of our study was only to determine and report the predictors of diabetes, we did not intend to compare the model with the crude model and, as our statistician recommended, we did not find it necessary to check the best fit model. 

Reviewer #2: All comments have been addressed. I got no more comments for the authors. They have also finish the English editing.

Thank you

References

[1] Agresti A. An introduction to categorical data analysis. 2 nd. John Wiley & Sons; 2018.

---

## [Decision Letter · Decision Letter 3]

28 Nov 2022

PONE-D-21-19349R3Gender and Urban–Rural Residency based Differences in the Prevalence of Type-2 DiabetesMellitus and its Determinants among Adults in Naghadeh: Results of IraPEN SurveyPLOS ONE

Dear Dr. Nadrian,

Thank you for submitting your manuscript to PLOS ONE. After careful consideration, we feel that it has merit but does not fully meet PLOS ONE’s publication criteria as it currently stands. Therefore, we invite you to submit a revised version of the manuscript that addresses the points raised during the review process.

ACADEMIC EDITOR: I have positive opinions about your article. We received minor revision from one reviewer and acceptance from another reviewer. The manuscript will be evaluated again after making corrections or presenting your explanations according to the reviewer's comments.

We look forward to receiving your revised manuscript.

Kind regards,

Ugurcan Sayili, M.D.

Academic Editor

PLOS ONE

Journal Requirements:

Reviewers' comments:

Reviewer's Responses to Questions

**Comments to the Author**

1. If the authors have adequately addressed your comments raised in a previous round of review and you feel that this manuscript is now acceptable for publication, you may indicate that here to bypass the “Comments to the Author” section, enter your conflict of interest statement in the “Confidential to Editor” section, and submit your "Accept" recommendation.

Reviewer #1: (No Response)

Reviewer #2: All comments have been addressed

2. Is the manuscript technically sound, and do the data support the conclusions?

Reviewer #1: Partly

Reviewer #2: Yes

3. Has the statistical analysis been performed appropriately and rigorously? 

Reviewer #1: Yes

Reviewer #2: Yes

4. Have the authors made all data underlying the findings in their manuscript fully available?

Reviewer #1: Yes

Reviewer #2: Yes

5. Is the manuscript presented in an intelligible fashion and written in standard English?

Reviewer #1: Yes

Reviewer #2: Yes

6. Review Comments to the Author

Reviewer #1: Thank you for your interesting manuscript. However, I have some comments yet.

1. Mercury sphygmomanometer was used for the measurement of blood pressure in the methodology. Actually, the digital blood pressure is recommended by WHO. Thus, it could be somewhat extent of variation in blood pressure and effect on analysis as I understand your availability of pressure cuff. However, you have to declare this variation and potential effect in your manuscript.

2. Fruit and vegetable consumption is one important factor for diabetes prevalence. However, this data is missing in your results.

3. Concerning odds ratio in table 3 and 4, you presented OR of age. I understand that you reported OR of age, but it seems to be you had used the continuous data for age by logistic regression. Thus, you could add the clarification.

4. In addition, you used the word “cholesterol” the table 3 & 4. Generally, we could use the word “total cholesterol” instead of cholesterol. Please use the relevant word while your data is fasting lipids.

Reviewer #2: All comments have been addressed. I got no more comments for the authors. This draft has also finished English editing.

7. PLOS authors have the option to publish the peer review history of their article (what does this mean?). If published, this will include your full peer review and any attached files.

Reviewer #1: No

Reviewer #2: No

---

## [Author Response · Author response to Decision Letter 3]

29 Nov 2022

Reviewer #1: Thank you for your interesting manuscript. However, I have some comments yet.

1. Mercury sphygmomanometer was used for the measurement of blood pressure in the methodology. Actually, the digital blood pressure is recommended by WHO. Thus, it could be somewhat extent of variation in blood pressure and effect on analysis as I understand your availability of pressure cuff. However, you have to declare this variation and potential effect in your manuscript.

Revisions:

Thanks for your valuable comments.

Discussion; limitations; paragraph 1; lines 12-16.

Despite the recommendation of WHO to use digital sphygmomanometer, mercury sphygmomanometer was used in the IraPen study, as digital blood pressure measuring device was not available in all rural and urban health centers of the country. So, it could be somewhat extent of variation in blood pressure, which might have some potential effect on the results.

2. Fruit and vegetable consumption is one important factor for diabetes prevalence. However, this data is missing in your results.

Revisions:

Discussion; limitations; paragraph 1; lines 21-22.

Fruit and vegetable consumption is an important factor for diabetes prevalence. However, this data is missing in our results.

3. Concerning odds ratio in table 3 and 4, you presented OR of age. I understand that you reported OR of age, but it seems to be you had used the continuous data for age by logistic regression. Thus, you could add the clarification.

Response: 

No, we used the categorical data for age. As Agresti recommended (1), Logistic regression, like ordinary regression, can have multiple explanatory variables, and some or all of those variables (as predictors) can be categorical, rather than quantitative. Also, an important interpretation of logistic regression model uses the odds ratio [1]. So, we can also use odds ratio as one of the ways to interpret such logistic regression analyses. 

[1] Agresti A. An introduction to categorical data analysis. 2 nd. John Wiley & Sons; 2018.

4. In addition, you used the word “cholesterol” the table 3 & 4. Generally, we could use the word “total cholesterol” instead of cholesterol. Please use the relevant word while your data is fasting lipids.

Revisions:

Tables 3 and 4 were revised, accordingly. 

Reviewer #2: All comments have been addressed. I got no more comments for the authors. This draft has also finished English editing.

Thank you

---

## [Decision Letter · Decision Letter 4]

19 Dec 2022

Gender and Urban–Rural Residency based Differences in the Prevalence of Type-2 DiabetesMellitus and its Determinants among Adults in Naghadeh: Results of IraPEN Survey

PONE-D-21-19349R4

Dear Dr. Nadrian,

We’re pleased to inform you that your manuscript has been judged scientifically suitable for publication and will be formally accepted for publication once it meets all outstanding technical requirements.

Kind regards,

Ugurcan Sayili, M.D.

Academic Editor

PLOS ONE

Additional Editor Comments (optional):

Reviewers' comments:

Reviewer's Responses to Questions

**Comments to the Author**

1. If the authors have adequately addressed your comments raised in a previous round of review and you feel that this manuscript is now acceptable for publication, you may indicate that here to bypass the “Comments to the Author” section, enter your conflict of interest statement in the “Confidential to Editor” section, and submit your "Accept" recommendation.

Reviewer #1: All comments have been addressed

Reviewer #2: All comments have been addressed

2. Is the manuscript technically sound, and do the data support the conclusions?

Reviewer #1: Yes

Reviewer #2: Yes

3. Has the statistical analysis been performed appropriately and rigorously? 

Reviewer #1: Yes

Reviewer #2: Yes

4. Have the authors made all data underlying the findings in their manuscript fully available?

Reviewer #1: Yes

Reviewer #2: Yes

5. Is the manuscript presented in an intelligible fashion and written in standard English?

Reviewer #1: Yes

Reviewer #2: Yes

6. Review Comments to the Author

Reviewer #1: Thank you for your revised version. I found you addressed the commnets. Therefore, I have no more suggestion or comments.

Reviewer #2: The authors have adequately addressed my comments raised in a previous round of review and I feel that this manuscript is now acceptable for publication.

7. PLOS authors have the option to publish the peer review history of their article (what does this mean?). If published, this will include your full peer review and any attached files.

Reviewer #1: No

Reviewer #2: No

---

## [Editor Report · Acceptance letter]

27 Feb 2023

PONE-D-21-19349R4 

Gender and Urban–Rural Residency based Differences in the Prevalence of Type-2 Diabetes Mellitus and its Determinants among Adults in Naghadeh: Results of IraPEN Survey 

Dear Dr. Nadrian:

I'm pleased to inform you that your manuscript has been deemed suitable for publication in PLOS ONE. Congratulations! Your manuscript is now with our production department. 

Kind regards, 

on behalf of

Dr. Ugurcan Sayili 

Academic Editor

PLOS ONE